# A receptor and neuron that activate a circuit limiting sucrose consumption

Ryan M Joseph[1], Jennifer S Sun[1], Edric Tam[2], John R Carlson[1]*

[1]Department of Molecular, Cellular and Developmental Biology, Yale University, New Haven, United States; [2]Department of Biomedical Engineering, Yale University, New Haven, United States

**Abstract** The neural control of sugar consumption is critical for normal metabolism. In contrast to sugar-sensing taste neurons that promote consumption, we identify a taste neuron that limits sucrose consumption in *Drosophila*. Silencing of the neuron increases sucrose feeding; optogenetic activation decreases it. The feeding inhibition depends on the IR60b receptor, as shown by behavioral analysis and $Ca^{2+}$ imaging of an *IR60b* mutant. The *IR60b* phenotype shows a high degree of chemical specificity when tested with a broad panel of tastants. An automated analysis of feeding behavior in freely moving flies shows that IR60b limits the duration of individual feeding bouts. This receptor and neuron provide the molecular and cellular underpinnings of a new element in the circuit logic of feeding regulation. We propose a dynamic model in which sucrose acts via IR60b to activate a circuit that inhibits feeding and prevents overconsumption.

## Introduction

Feeding regulation is a critical problem in animal life (*Morton et al., 2006*). Organisms must consume sufficient levels of nutrients to thrive, but overconsumption can have severe consequences. The initiation of feeding has been studied in the model genetic system *Drosophila,* whose taste system has many parallels to that of mammals (*Liman et al., 2014*). A variety of molecular and cellular mechanisms must operate for feeding to be initiated. Discrete classes of taste receptors and neurons assess the molecular composition of a potential food source (*Marella et al., 2006*; *Thorne et al., 2004*). Receptors sensitive to sugars signal the presence of nutrients (*Dahanukar et al., 2007*; *Freeman et al., 2014*; *Jiao et al., 2008*; *Wisotsky et al., 2011*); receptors sensitive to bitter-tasting compounds signal the danger of toxins (*Lee et al., 2009*, *2015*; *Shim et al., 2015*; *Weiss et al., 2011*). If nutrient levels are sufficiently high and toxin levels sufficiently low, the animal begins to feed (*French et al., 2015*).

The termination of feeding, once begun, is poorly understood. When nutrients are readily available and toxins are absent, what mechanisms terminate feeding? Previously described mechanisms that terminate feeding are based in central neural circuits that act downstream from taste neurons (*Hergarden et al., 2012*; *Pool et al., 2014*). Some involve internal sensors that monitor post-ingestive nutrient concentrations in the hemolymph (*Dus et al., 2015*; *Miyamoto et al., 2012*) or mechanical tension in the gut (*Olds and Xu, 2014*), while others involve cells in the taste center of the brain that are controlled by the satiety state of the fly (*Marella et al., 2012*; *Yapici et al., 2016*; *Zhan et al., 2016*).

It would seem advantageous for an animal to have an additional means of inhibiting feeding, a mechanism that operates on a fast time-scale and functions directly in the gustatory organs. Such a mechanism could prevent overconsumption at an early stage, before the animal has invested in the ingestion of nutrients that may be not only unnecessary, but detrimental.

*For correspondence: john.carlson@yale.edu

Competing interests: The authors declare that no competing interests exist.

**eLife digest** All animals – from the fruit fly to mammals like humans – must control their dietary intake of nutrients to survive and stay healthy. Taste receptors that sense high-calorie sugars are essential to this process. Typically, when food tastes sweet, it signals that the food contains nutrients and promotes consumption. However, eating too much sugar can be detrimental because the animal wastes time and energy eating food that it does not need, and could eventually lead to obesity and other metabolic diseases. This raised the question: are there any taste receptors that, once they detect sugars, cause animals to eat less?

Joseph et al. worked with the fruit fly *Drosophila melanogaster* and identified one such taste receptor called IR60b. The experiments showed that this taste receptor responds selectively to sucrose (a high-calorie sugar), and that it activates nerve cells that cause fruit flies to eat less food, rather than more. When the receptor was experimentally inactivated, the fruit flies ate for longer and ate too much sucrose. This indicates that the flies need this receptor to control their sugar intake.

A next step will be to see if mammals similarly use sweet-sensing taste receptors to limit the amount of food they eat. A better insight into how mammals can control what they eat could provide a deeper understanding of how to tackle major health issues, such as obesity, in humans.

The gustatory organs of the fly include the legs, the labellum, and the pharyngeal sense organs, which include the labral sense organ (LSO)(*Gendre et al., 2004*; *Stocker, 1994*). Taste reception in these organs is mediated by a large number of receptors, including those of the Gustatory receptor (Gr) family (*Clyne et al., 2000*; *Scott et al., 2001*), which detect a variety of sugars and bitter compounds (*Liman et al., 2014*). Members of an ancient class of sensory receptors called the ionotropic receptors (IRs) were recently found be expressed in taste organs (*Benton et al., 2009*; *Croset et al., 2010*). In particular, a large clade of IRs called the IR20a clade were mapped to taste neurons in the legs, labellum, and pharynx (*Koh et al., 2014*; *Stewart et al., 2015*). The functions of the *IR20a* clade genes are virtually unexplored, although analysis of two that are expressed in the male leg has revealed roles in male mating behavior, presumably as pheromone receptors (*Koh et al., 2014*).

Among the ~35 members of the *IR20a* clade, one gene, *IR60b,* is exceptional in two ways. First, it is remarkably restricted in its expression, as determined by a systematic analysis of *IR-GAL4* drivers (*Koh et al., 2014*). Expression was detected in the pharynx, in a single pair of neurons, and nowhere else in the animal. Second, *IR60b* is unique among the genes of this *IR* clade in showing the signature of purifying selection (*Figure 1D*) (*Koh et al., 2014*; *Mackay et al., 2012*; *Stoletzki and Eyre-Walker, 2011*), suggesting that *IR60b* represents a particularly effective solution to a difficult evolutionary problem.

Here, we show that the pair of neurons in which IR60b is expressed mediates a response to sucrose. Strikingly, these IR60b neurons inhibit sucrose consumption, in contrast to other sugar-sensitive taste neurons that promote sucrose consumption. Silencing the IR60b neurons increases sucrose uptake, while activating these neurons reduces sugar uptake. Sucrose detection by these neurons depends on the IR60b receptor, as revealed in behavioral and Ca$^{2+}$ imaging analysis of an *IR60b* mutant. The *IR60b* phenotype shows a high degree of specificity when tested with a broad panel of tastants. An automated analysis of feeding behavior in freely moving flies shows that IR60b acts to limit the duration of feeding bouts. Taken together, this study identifies a novel function for an understudied class of receptor and taste neuron. The analysis reveals a new element in the circuit logic of feeding regulation: sugar-sensing taste neurons that act to prevent overconsumption.

## Results

### *IR60b* is expressed in an orphan neuron of the pharynx

We sought to determine whether IR60b is expressed in neurons that had previously been defined, as opposed to in 'orphan' neurons. IR60b is expressed in the LSO in sensillum 7 (*Figure 1A–C*), which houses eight neurons (*Gendre et al., 2004*). Two of these neurons have been shown to

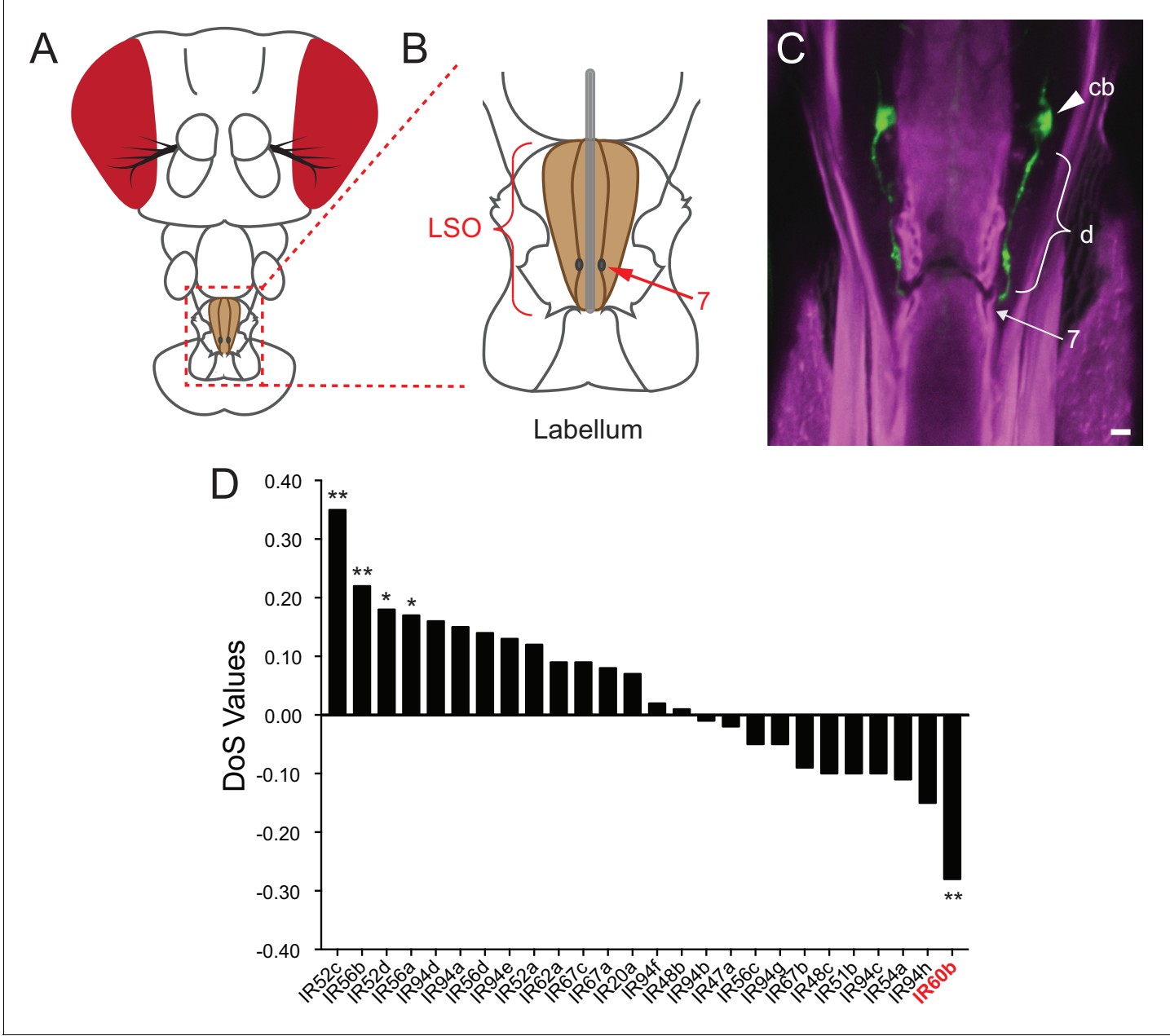

**Figure 1.** Expression of IR60b in the LSO. (**A**) *Drosophila* head. The box indicates the region of the proboscis containing the labral sense organ (LSO, shaded) of the pharynx. (**B**) The pharyngeal region containing the LSO. The position of sensillum 7 is indicated. (**C**) *IR60b-GAL4; UAS-GFP* shows expression in a single pair of neurons that project dendrites (d) to the pore of sensillum 7, whose position is indicated. (cb), cell body. To maximize the fidelity of the driver, *GAL4* was placed between sequences lying 5' and 3' to *IR60b*. Scale bar = 5 µm. Green color represents *UAS-GFP* fluorescence, visualized with a 488 nm laser. Magenta color represents cuticular autofluorescence, visualized with a 514 nm laser. (**D**) *IR60b* is the only member of the *IR20a* clade that shows a significantly negative Direction of Selection (DoS) signature. Values were generated by using polymorphism data from the Drosophila Genetic Reference Panel (*Huang et al., 2014*; *Mackay et al., 2012*) to perform McDonald-Kreitman Tests (*Stoletzki and Eyre-Walker, 2011*) for the IR20a clade genes. Values were generated using the popDrowser website (*Ràmia et al., 2012*). Briefly, the values are calculated by comparing sequence variation within *Drosophila melanogaster* to the sequence divergence between *Drosophila melanogaster* and its sibling species *Drosophila simulans*. *p<0.05; **p<0.01. Figure adapted from data displayed in *Figure 6K* in *Koh et al. (2014)*.

coexpress a set of sugar-sensing *Gr-GAL4 or Gr-LexA* drivers, including drivers of *Gr43a*, *Gr61a*, *Gr64a*, *Gr64e*, and *Gr64f* (*LeDue et al., 2015*). These two neurons promote the consumption of

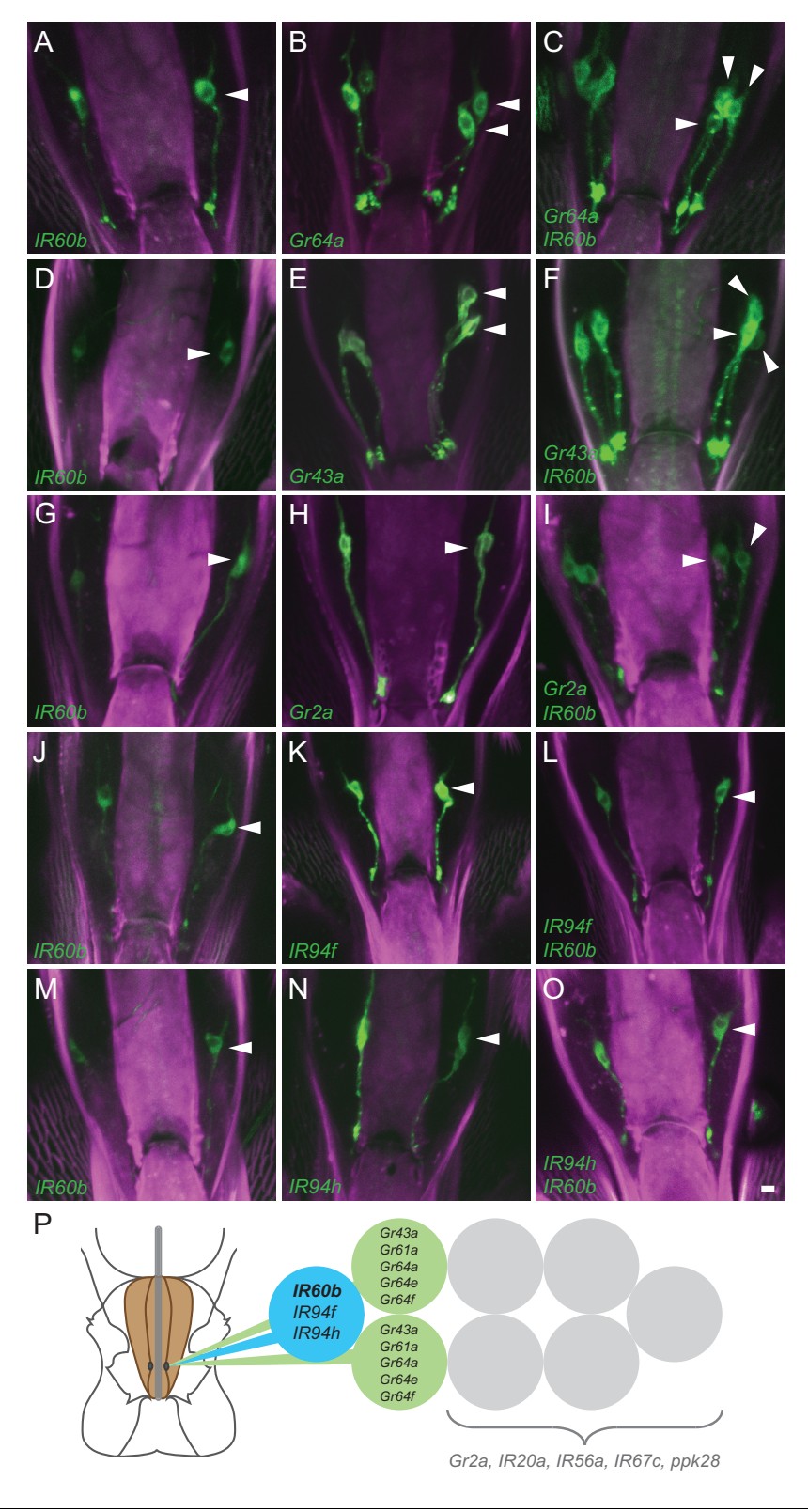

**Figure 2.** Coexpression of *IR60b* and other receptors. (A–O) Maximum intensity projections of *GFP* expression in *IR60b-GAL4/+; IR60b-GAL4/UAS-mCD8-GFP* flies (A,D,G,J,M); *Gr-* or *IR-GAL4/+; UAS-mCD8-GFP/+* flies (B,E,H,K,N); *IR60b-GAL4/Gr-* or *IR-GAL4; IR60b-GAL4/UAS-mCD8-GFP* flies (C, F,I,L,O). Two copies of *IR60b-GAL4* were used to compensate for the low expression level of *IR60b-GAL4*. Arrowheads indicate neuronal cell bodies in the LSO. Scale bar = 5 µm. (P) Mapping of *IR60b* driver relative to other *IR* and *Gr* drivers in the eight gustatory neurons of LSO sensillum 7. Drivers

*Figure 2 continued on next page*

*Figure 2 continued*

mapped to neurons indicated in color do not co-express with drivers mapped to neurons indicated in gray, which have not been mapped at single-cell resolution in this analysis. *Gr66a* is not included in the diagram because the *Gr66a-GAL4* driver maps to other sensilla of the LSO.

The following figure supplement is available for figure 2:

**Figure supplement 1.** *IR60b-GAL4* expresses in a neuron of the pharynx that does not express several other drivers.

sweet foods. The other six neurons have not been assigned a molecular identity, and their function is unknown.

To determine whether *IR60b* is expressed in an orphan neuron, we performed a systematic double-driver analysis. We first combined *IR60b-GAL4* with *Gr-GAL4* drivers and used *UAS-GFP* (*Lee and Luo, 1999*) to count the number of labeled cells in the LSO. The neurons in this sensillum are tightly grouped, but careful analysis of the confocal z-stacks allowed us to determine the number of labeled neurons. We found that while *IR60b-GAL4* labels a single neuron in the LSO (*Figure 2A*; the arrowhead indicates a labeled cell body in one of the two bilaterally symmetric neurons), and *Gr64a-GAL4* labels two neurons (*Figure 2B*), together the two drivers label three neurons (*Figure 2C*). The simplest interpretation of these results is that *IR60b* is expressed in a neuron distinct from those that express *Gr64a*. Similar analysis with the *Gr43a-GAL4* driver confirmed this conclusion (*Figure 2D–F*).

We next tested other *Gr* drivers that are expressed in the *Drosophila* pharynx (*Joseph and Heberlein, 2012*; *Kwon et al., 2014*). The labeling pattern of *IR60b-GAL4* is again distinct from that of *Gr2a-GAL4* (*Figure 2G–I*), and that of *Gr66a-GAL4* (*Figure 2—figure supplement 1A–C*).

Although the *IR60b* driver did not coexpress with any of these *Gr* drivers, it did coexpress with *IR94f-GAL4* (*Figure 2J–L*) and *IR94h-GAL4* (*Figure 2M–O*). We also tested *IR67c, IR20a*, and *IR56a* drivers and found that all three are expressed in this sensillum, but in neurons distinct from that expressing *IR60b-GAL4* (*Figure 2—figure supplement 1D–L*). Likewise, a driver representing ppk28, an osmosensitive ion channel that mediates water consumption (*Cameron et al., 2010*; *Thistle et al., 2012*), is expressed in the sensillum but in a different neuron (*Figure 2—figure supplement 1M–O*).

Taken together, this mapping analysis suggests that IR60b is expressed in a previously undefined gustatory neuron of the *Drosophila* pharynx. The analysis also suggests that IR60b is coexpressed with two other IRs of the IR20a clade (*Figure 2P*).

## Silencing of the IR60b neuron increases feeding on sucrose

We asked whether the IR60b neuron acts in the regulation of feeding, given its location in a sensillum of the pharynx. We used a modified pharyngeal pumping paradigm that combines elements of previously described cibarial pumping assays (*Figure 3A,B*) (*Manzo et al., 2012*; *Qi et al., 2015*). Our paradigm allows measurement of several parameters of feeding over a short time scale, that is on the order of a minute. The parameters include total feeding time, swallowing rate, and number of swallows. To facilitate analysis of food consumption by direct visual inspection, we added a blue dye to the food samples. Control experiments indicated that the presence of the dye did not affect the results (*Figure 3—figure supplement 1*).

As food samples, we initially used drops of sucrose solution. We verified that flies exhibit robust consumption of sucrose solutions in this paradigm across a wide range of concentrations (*Figure 3C*). The total time of feeding increased with sucrose concentration and saturated at ~100 mM. The rate of pumping in this paradigm, that is the swallowing rate, was independent of concentration (*Figure 3D*), consistent with previous results that pumping rate is not impeded by viscosity at sucrose concentrations less than 1M (*Manzo et al., 2012*). To confirm the constant pumping rate, we plotted the number of swallows against total time of feeding for each fly, pooled across all sucrose concentrations and found a strong correlation (*Figure 3E*, $R^2$ = 0.91; p<0.001). To determine whether the time of feeding in turn correlated with the volume of food ingested, we made extracts from individual flies after recording the time of feeding. We measured the light absorbance of each extract to measure the amount of blue dye consumed, and from this value calculated the

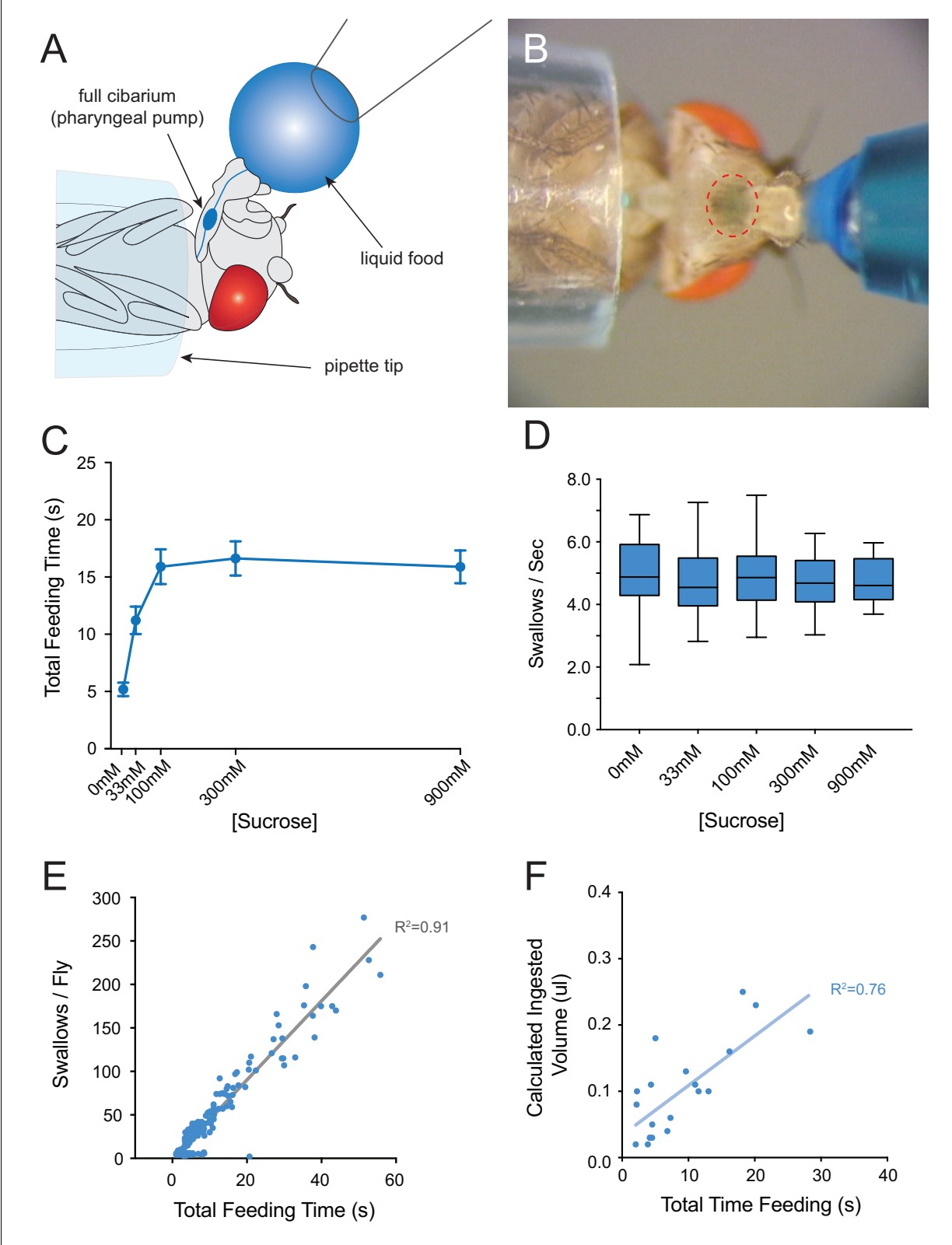

**Figure 3.** Consumption of sucrose in a modified pharyngeal pumping assay. (**A**) Diagram of assay. Figure adapted from *Figure 2B* of *Delventhal et al. (2017)*. (*Delventhal et al., 2017*) (**B**) Blue dye allows visualization of filling and emptying of the cibarium (red-dashed circle). (**C**) Total feeding time of w Canton-S females as a function of sucrose concentration. n = 37–108. (**D**) Swallowing rate of flies as a function of sucrose concentration. Medians do not differ (one-way ANOVA, Bonferroni post-test, n = 22–34). Lines indicate medians; boxes indicate 25% quartiles above and below the median; whiskers

*Figure 3 continued on next page*

*Figure 3 continued*

indicate range of values. (E) Correlation analysis of total number of swallows per fly vs. total feeding time. Data are from all concentrations tested in (C). $R^2 = 0.91$, ***p<0.001, Pearson's correlation test, n = 140. (F) Correlation analysis of calculated volumes of 900 mM sucrose ingested vs. total feeding time (***p<0.001, $R^2 = 0.76$, Pearson's correlation test, n = 20). Flies consumed ~8 nanoliters/s, which is comparable to reported values (*Qi et al., 2015*; *Yapici et al., 2016*). Volumes were determined by extracting dye from single flies after feeding, and measuring the absorbance of each sample. Absorbance values were converted into calculated volumes using the slope of a standard absorbance curve for the concentration of the blue dye (see Materials and methods). 12 hr starved, mated females were used in this and all other assays.

The following figure supplement is available for figure 3:

**Figure supplement 1.** Addition of erioglaucine blue dye to sucrose food does not affect feeding behaviors.

volume of sucrose solution consumed (Materials and methods). The feeding time was in fact correlated with the volume ingested (*Figure 3F*, $R^2 = 0.76$; p<0.001). Thus, the total feeding time serves as a reasonable measure of the amount of consumption in this paradigm.

To test whether the IR60b neuron acts in feeding regulation, we used *IR60b-GAL4* to drive *UAS-tetanus toxin (TNT)*, which blocks synaptic signaling from neurons (*Martin et al., 2002*; *Sweeney et al., 1995*). We compared feeding time of *IR60b-GAL4/UAS-TNT* flies to those of parental *IR60b-GAL4* and *UAS-TNT* control flies. Feeding time was scored blind to genotype. As a precaution against genetic background effects, the *IR60b-GAL4* and *UAS-TNT* constructs were each backcrossed for at least five generations against our control genetic background before the experiment.

Surprisingly, silencing of IR60b neurons caused a robust increase in feeding time when flies were tested with 300 mM sucrose (*Figure 4A*). We confirmed and extended this finding by testing across a broad range of sucrose concentrations. At all sucrose concentrations, the silenced flies showed greater feeding times than the parental controls, which were equivalent to each other (*Figure 4B*). Feeding time was equivalently low among the three genotypes when tested with a control water solution, arguing that the silencing of IR60b neurons did not affect baseline levels of thirst or satiety. The silencing of the IR60b neuron did not affect pumping rate at any concentration (*Figure 4C*), arguing that increased feeding times were not due to motor defects that caused flies to swallow at a lower rate. The simplest interpretation of these results is that the IR60b neuron acts to limit the intake of the sucrose source.

To confirm the role of the IR60b neuron, we used an independent driver, *IR94f-GAL4,* which we had found to be coexpressed with *IR60b-GAL4* in the same neuron (*Figure 2L*). We confirmed with this driver that silencing the neuron increased the intake of the sucrose source (*Figure 4—figure supplement 1*).

Does this neuron affect intake of sugars other than sucrose? We tested 300 mM concentrations of fructose, trehalose, and glucose, and found no effects of silencing (*Figure 4D*). This specificity suggests that the neuron is tuned to a relatively narrow subset of taste stimuli, a hypothesis that is considered further below.

## Activation of the IR60b neuron decreases feeding time

Since silencing the IR60b neuron increased feeding, we wondered if activating the neuron would decrease feeding. To activate the IR60b neuron, we used *UAS-CsChrimson,* which encodes a channelrhodopsin that is activated by long-wavelength red light (*Klapoetke et al., 2014*). As an initial test, we asked whether optogenetic activation of the IR60b neuron with red light would decrease sucrose consumption. We found that it did not (*Figure 4—figure supplement 2*); however, it seemed likely that the sucrose stimulus alone may have activated the neuron to such an extent that additional activation by optogenetics would not produce a detectable effect. Further analysis shown below confirmed that sucrose activates the IR60b neuron directly (Figure 7F,G). We therefore wished to test a stimulus that did not activate the IR60b neuron but that elicited a substantial baseline feeding level. Accordingly, we used trehalose, which elicits a substantial level of feeding, and whose consumption was not affected by silencing of the IR60b neuron (*Figure 4D*); further analysis shown below confirmed that trehalose does not activate the IR60b neuron (Figure 7F,G). Thus, we asked

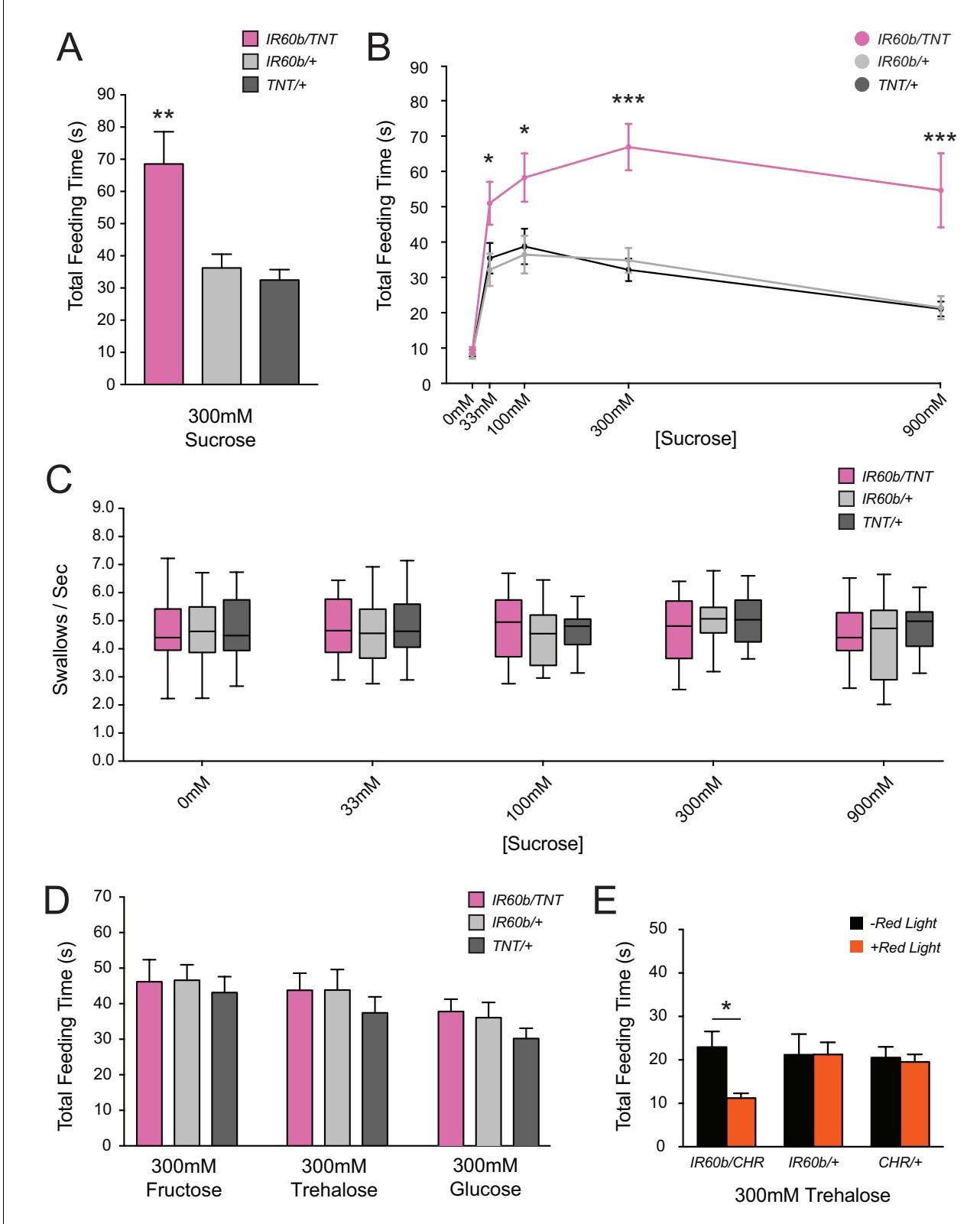

**Figure 4.** Silencing and activating the *IR60b* neuron alters feeding time. (**A**) Initial analysis of total feeding time of *IR60b-GAL4/UAS-TNT*, *IR60b-GAL4/+*, and *UAS-TNT/+* flies when 300 mM sucrose was delivered in the pharyngeal pumping assay (**p<0.01, one-way ANOVA, Bonferroni post-test, n = 16–24). (**B**) Total feeding time of silenced flies at a range of sucrose concentrations (*p<0.05, ***p<0.001: two-way ANOVA, Bonferroni post-test, n = 18–30). (**C**) Pumping rates of silenced flies at different sucrose concentrations. No differences were observed in pumping rates among different

*Figure 4 continued on next page*

*Figure 4 continued*

genotypes or doses (two-way ANOVA, Bonferroni post-test, n = 16–27). (D) Total feeding time of silenced flies when tested with glucose, fructose, or trehalose. No increases in sugar consumption were observed in *IR60b-GAL4/UAS-TNT* flies when compared to controls (one-way ANOVA, Bonferroni post-test, n = 25–38). (E) *UAS-Chrimson* (*CHR*) activation: total feeding time with 300 mM trehalose in *IR60b-GAL4/UAS-Chrimson*, *IR60b-GAL4/+*, and *UAS-Chrimson/+* flies under low-light conditions, either with or without application of red light. *IR60b-GAL4/UAS-Chrimson* flies show a decrease in feeding when *IR60b* neurons are activated with red light (*p<0.05, two-way ANOVA, Bonferroni post-test, n = 12–24). Under red light, feeding time for *IR60b-GAL4/UAS-Chrimson* flies is lower than for *IR60b-GAL4/+* and *UAS-Chrimson/+* controls (*p<0.05, two-way ANOVA, Bonferroni post-test, n = 12–24).

The following figure supplements are available for figure 4:

**Figure supplement 1.** Expression of *UAS-TNT* driven by *IR94f-GAL4* also causes overconsumption.

**Figure supplement 2.** Optogenetic activation of IR60b neurons does not reduce feeding on sucrose.

whether trehalose feeding would be decreased when the IR60b neuron was activated optogenetically.

We found that optogenetic activation of the IR60b neuron with red light reduced trehalose feeding of *IR60b-GAL4/UAS-Chrimson* flies (*Figure 4E*). As a control, red light did not affect the feeding of either parental genotype. Moreover, in the absence of red light, all genotypes showed the same level of consumption. These results indicate that activation of IR60b neurons is sufficient to reduce trehalose consumption. In summary, the activity of IR60b neurons is necessary to limit sucrose consumption to control levels (*Figure 4A,B*) and is sufficient to limit consumption of another sugar (*Figure 4E*).

Taken together, the silencing and activation results reveal a surprising role for the IR60b neuron, which appears to negatively regulate ingestion in response to sucrose, a nutritive and appetitive sugar that normally promotes feeding.

## The IR60b receptor acts in limiting sucrose consumption

Our results have shown that the activity of the IR60b neuron limits sucrose consumption. Is the activity of the IR60b receptor required for this limitation? *IR60b, IR94f,* and *IR94h* drivers are coexpressed in this neuron, and *a priori* any of them, or none of them, could act in the regulation of sucrose feeding.

To test the hypothesis that the IR60b receptor functions in the limitation of sucrose feeding, we used CRISPR-Cas9 to create a mutant allele that lacks 66% of the *IR60b* coding sequence (*Gratz et al., 2014*) (Materials and methods). The mutation was designed to eliminate all three of the predicted three transmembrane domains of IR60b. We confirmed the presence of the mutation, *IR60b[1]*, by PCR analysis (*Barnes, 1994*). The cleavage sites chosen for the construction of the deletion had no predicted off-target sites. We backcrossed the mutation for five generations against our control genetic background prior to testing.

The *IR60b[1]* mutant showed a robust increase in feeding time when tested with 300 mM sucrose (*Figure 5A*). The homozygous *IR60b[1]/IR60b[1]* mutant showed a much greater feeding time than either the parental control or an *IR60b[1]/+* heterozygote. The heterozygote showed the same phenotype as the control, supporting the interpretation that the phenotype is due to a recessive mutation in *IR60b*. We confirmed and extended these results by testing a broad range of sucrose concentrations (*Figure 5B*). At all sucrose concentrations, the mutant showed greater feeding times than the controls. When tested with a control water drop, feeding time was equivalently low among all three genotypes, arguing that increased feeding times in the mutant did not result from differences in baseline levels of thirst.

As a further confirmation of the requirement for *IR60b*, we tested an independent allele, *IR60b[2]*, which was generated and backcrossed using the same protocol as for *IR60b[1]*. This second allele produced an increase in feeding time very similar to that of *IR60b[1]* (*Figure 5C,D*).

The greater feeding time of the mutant could reflect greater food consumption. Alternatively, in principle the mutant could consume the same amount as the controls if the greater feeding time of the mutant were a mechanism of compensating for a decreased feeding rate. To distinguish

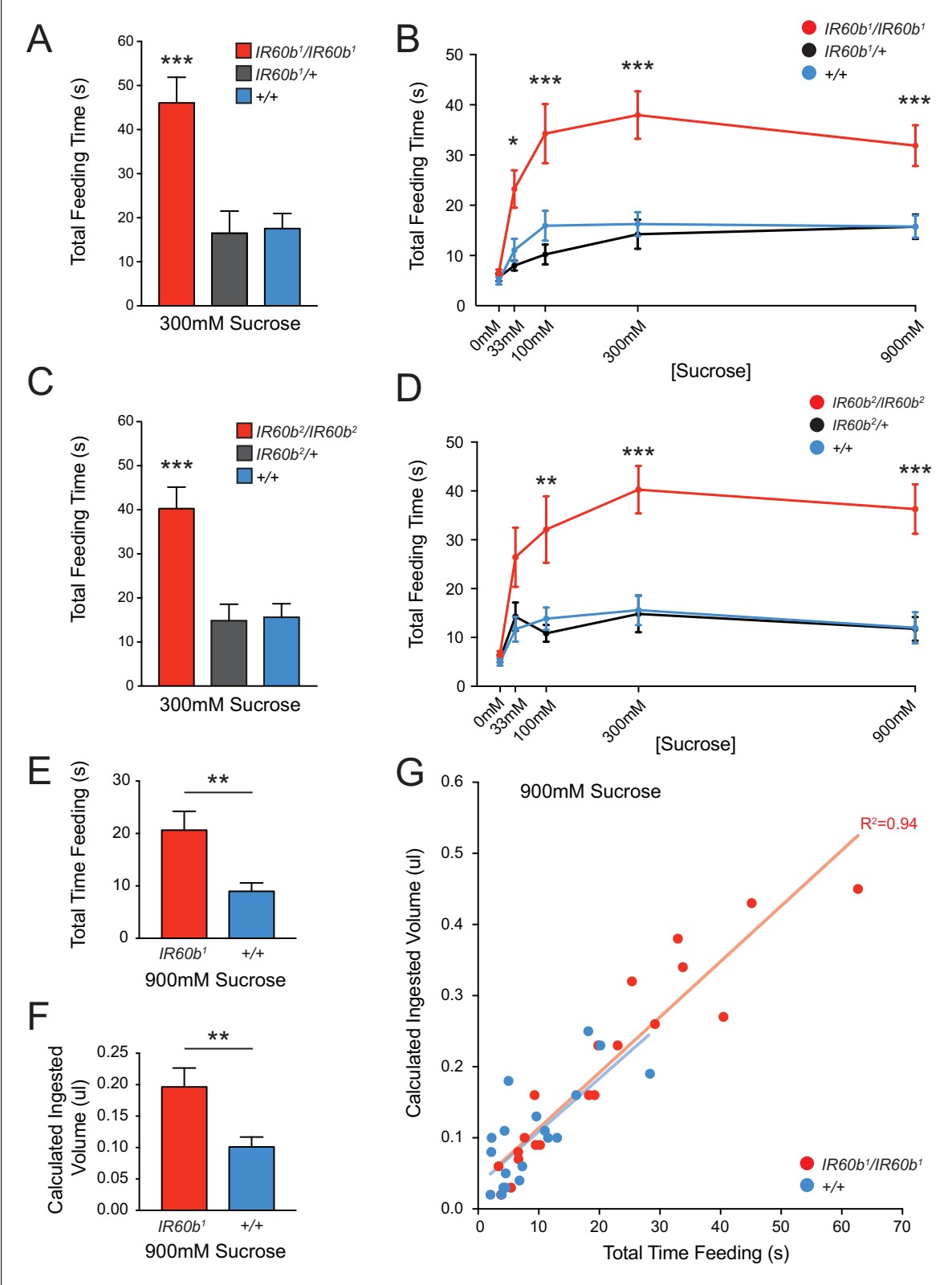

**Figure 5.** Mutation of *IR60b* leads to overconsumption of sucrose . (**A**) Initial analysis of feeding time for *IR60b¹/IR60b¹*, *IR60b¹/+*, and *w Canton-S (+/+)* flies when 300 mM sucrose was delivered in the pharyngeal pumping assay (***p<0.001, one-way ANOVA, Bonferroni post-test, n = 13–20). (**B**) Sucrose dose-response curve for *IR60b¹* (*p<0.05, ***p<0.001, two-way ANOVA, Bonferroni post-test, n = 12–25 for 0 mM, n = 30 for other doses). We note that the feeding times in this experiment are somewhat less than those in *Figure 4B*, which may reflect differences in genetic background: all flies used in

*Figure 5 continued on next page*

*Figure 5 continued*

**Figure 5B** are *w*[1118], whereas all flies in **Figure 4** have a *mini-white*[+] marker in the *GAL4* and *UAS* constructs. (**C**) Initial analysis of *IR60b*[2] (***p<0.001, one-way ANOVA, Bonferroni post-test, n = 20–21. (**D**) Sucrose dose-response curve for *IR60b*[2] (**p<0.01, ***p<0.001, two-way ANOVA, Bonferroni post-test, n = 12–25). (**E**) The total time feeding and (**F**) calculated volume ingested for *IR60b*[1]/*IR60b*[1] and *+/+* flies when offered 900 mM sucrose, a concentration selected to elicit robust feeding. (**G**) Volume consumed versus total feeding time for individual flies. Volumes ingested were calculated as described in **Figure 3F**. Data for *+/+* shown here are from **Figure 3F** and are presented for comparison. The volume consumed correlated with total feeding time for both *IR60b*[1]/*IR60b*[1] and *+/+* flies (***p<0.001, $R^2$ = 0.94 and 0.76, respectively, Pearson's correlation test, n = 20). Slopes of regression lines for *IR60b*[1]/*IR60b*[1] and *+/+* were very similar (7.8 and 7.5 nl/s, respectively), showing that both mutant and control flies swallowed the same amount of food per unit time, and arguing that the increased feeding times observed in the mutant translate directly to increased volumes ingested.

between these two possibilities, we compared the quantity of food consumed by individual mutant flies and, in parallel, individual control flies. For each fly, we measured feeding time when provided a drop of 900 mM sucrose. We subsequently determined the volume ingested by measuring the amount of blue dye present in extracts of each individual fly. We found that not only was the mean feeding time greater for the mutant than the control (**Figure 5E**), but the volume ingested was greater as well (**Figure 5F**). The mutant and control flies showed the same linear relationship between feeding time and volume consumed; the slopes were statistically indistinguishable, consistent with an equivalent feeding rate (**Figure 5G**). The simplest interpretation of these results is that the *IR60b* mutant has a defect in the regulation of sucrose consumption, and that the IR60b receptor acts in limiting the intake of the sucrose source.

## The IR60b receptor regulates feeding of a highly specific set of taste stimuli

We were interested in the specificity of IR60b: does it act to restrict feeding of many stimuli, or just sucrose? We initially addressed this question in vivo, via a behavioral analysis.

First, we analyzed three classes of sugars and sugar alcohols: (i) sugars that *Drosophila* encounters in its natural environment that are highly appetitive in a variety of taste assays, and have high-caloric value (sucrose, glucose, fructose, trehalose, glycerol, maltose) (**Amrein and Thorne, 2005**; **Gordesky-Gold et al., 2008**); (ii) a sugar alcohol that is less appetitive but has high-nutritional value (sorbitol) (**Stafford et al., 2012**); (iii) sugars that are highly appetitive but have limited caloric value (arabinose and L-fucose) (**Stafford et al., 2012**). Sugars were offered at 100, 300, and 900 mM concentrations. We note that some of these stimuli have viscosities comparable to that of sucrose (**Galmarini et al., 2011**; **Nikam et al., 2000**; **Sheely, 1932**; **Swindells et al., 1958**).

Only sucrose, among the nine sugars tested, elicited increased feeding in the *IR60b*[1] mutant at concentrations of either 100 mM or 300 mM (**Figure 6A,B**). At 900 mM concentrations, sucrose and glucose elicited increased feeding (**Figure 6C**). The mutation had no effect on responses to lower concentrations of glucose, consistent with the lack of an effect on response to 300 mM glucose when silencing the IR60b neuron (**Figure 4D**). The feeding times of all other sugars were unaffected at all concentrations.

Bitter compounds were also tested. Bitter compounds inhibit the sugar responses elicited by many taste neurons (**French et al., 2015**; **Sellier et al., 2011**), although some taste neurons are activated by both classes of tastants (**van Giesen et al., 2016a**). We wondered if bitter compounds inhibited sugar response in this system and if so whether the inhibition occurred via *IR60b*. We tested each of several bitter compounds together with 300 mM trehalose, which was selected because the response to 300 mM trehalose alone is not affected by *IR60b*[1] (**Figure 6B**).

We tested a series of concentrations of three bitter compounds: caffeine, coumarin, and lobeline, and found that in wild type all three reduce mean trehalose feeding time, in a dose-dependent fashion (**Figure 6D**). The extent of reduction was not affected by the *IR60b* mutation.

Salts and pH were also tested since IR-expressing neurons in *Drosophila* chemosensory systems have been found to detect these stimuli (**Ai et al., 2013**; **Hussain et al., 2016**; **Zhang et al., 2013**). We found no effect of *IR60b*[1] on the feeding time elicited by a range of NaCl or KCl concentrations (**Figure 6E**). Nor was there an effect on feeding at a range of pH values. Finally, we tested several amino acids and found no effect of *IR60b*[1] (**Figure 6F**).

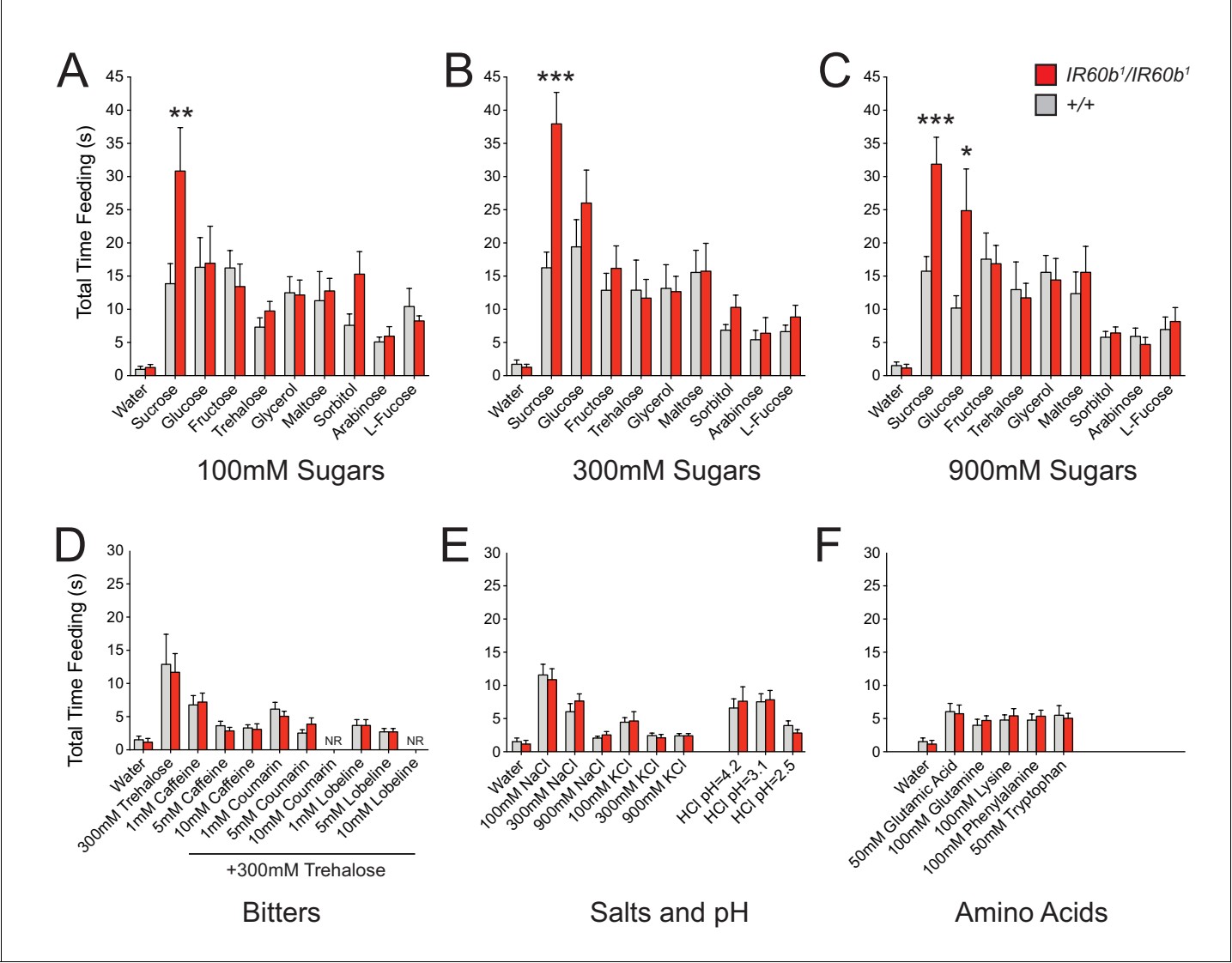

**Figure 6.** The IR60b receptor regulates feeding of a highly specific set of taste stimuli . (A–F) Total feeding times of *IR60b[1]/IR60b[1]* and *w Canton-S (+/+)* females with sugars, bitter compounds, salts, acids, and amino acids. Bitter stimuli, including all doses of caffeine, were delivered in a 300 mM trehalose background to stimulate baseline consumption to a level where decreases in feeding could be observed. All other compounds were dissolved in water. NR indicates flies for 10 mM coumarin and 10 mM lobeline groups did not accept these tastants. Differences between the mutant and *+/+* were observed only for 100, 300, and 900 mM sucrose, and 900 mM glucose (*p<0.05, **p<0.01, ***p<0.001, Student's two-tailed t-test, n = 10–32).

In summary, testing with a broad panel of taste stimuli revealed that *IR60b[1]* affected the feeding time only with sucrose and with a high concentration of glucose. These results suggested the hypothesis that the IR60b receptor responds to sucrose with a high degree of specificity.

## The IR60b neuron responds to sucrose and the response depends on IR60b

The behavioral analysis we have performed with flies in which the IR60b neuron is silenced, and with flies in which the IR60b receptor is mutated, together support a model in which the IR60b neuron is a sucrose sensor, and that IR60b is required for the neuron to detect sucrose. To test this model, we used calcium imaging to measure directly the response of the neuron to sucrose.

Previous imaging of sugar-sensitive pharyngeal neurons that express *Gr43a* drivers has focused on their axonal projections in the brain, visualized by cutting a window in the head (**LeDue et al.,**

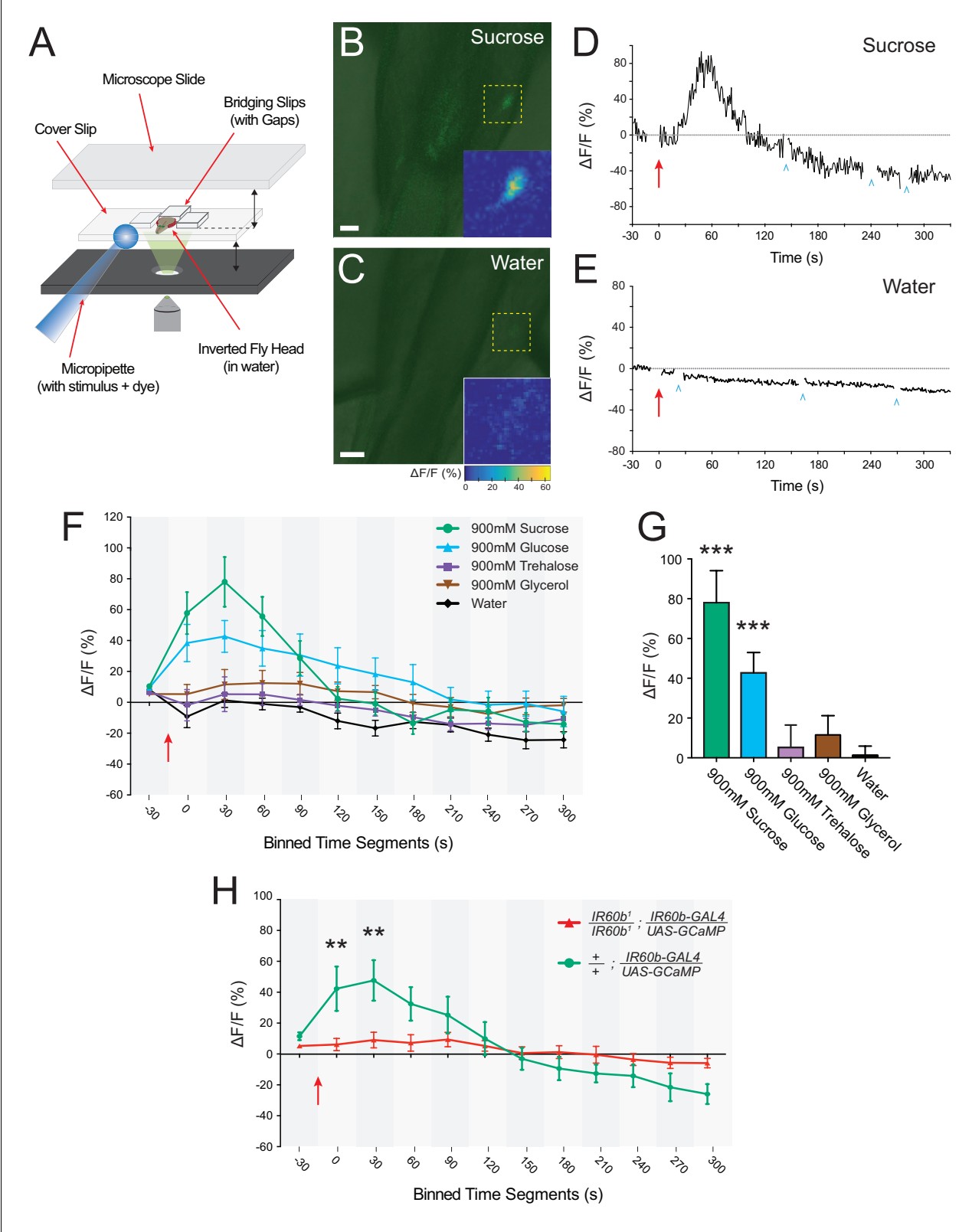

**Figure 7.** The IR60b neuron responds to sucrose and the response depends on IR60b . (**A**) The preparation used for imaging of cell bodies of the IR60b neurons. (**B–C**) Confocal images of merged green fluorescence and DIC channels showing *IR60b-GAL4/+; IR60b-GAL4/UAS-GCaMP6s* preparations during the delivery of (**B**) 900 mM sucrose or (**C**) water. Inset heat maps show the change in fluorescence at 60 s, after the addition of the stimulus. Yellow dashed boxes outline the regions shown in heat maps. Scale bars = 10 um. (**D–E**) Representative traces showing changes in fluorescence for (**D**)

*Figure 7 continued on next page*

*Figure 7 continued*

the 900mM sucrose stimulus or (**E**) water. Blue arrowheads indicate frames excluded because the sample was being re-focused to the appropriate plane of focus. Red arrows indicate time of stimulus application. (**F**) Time course of change in fluorescence ($\Delta$F/F) for flies that received the indicated stimuli. Values on x-axis represent binned, 30 s intervals, for example, "0" indicates the 0-29 s bin; "30" indicates the 30-59 s bin. The 900mM sucrose response differed from the water response between 0 and 120 s ($p^{***}<0.001$ for 0-29 s, 30-59 s, and 60-89 s bins; $^*p<0.05$ for the 90-119 s bin, two-way ANOVA, Bonferroni post-test, n=7-12). The response to the 900 mM glucose stimulus differed from the water response between 0 and 180 s ($p^{***}<0.001$ for 0-29 s and 30-59 s bins, and $^{**}p<0.01$ for 60-89 s, 90-119 s, 120-149 s, and 150-179 s bins, n=7-12). (**G**) Bar graph of $\Delta$F/F values from the 30-59 s bin in (**F**), illustrating differences during the bin of maximal fluorescence. (**H**) Time course of fluorescence in *IR60b[1]/IR60b[1]; IR60b-GAL4/UAS-GCaMP6s* flies and control flies in response to 900mM sucrose. Values on x-axis are binned in 30 s intervals as in (**F**). $\Delta$F/F values differed between 0 and 60 s ($^{**}p<0.01$ for 0-29 s and 30-59 s bins, two-way ANOVA, Bonferroni post-test, n=7-12).

The following figure supplement is available for figure 7:

**Figure supplement 1.** Response to sucrose in a neuron in which *UAS-GCaMP6s* is driven by *IR94f-GAL4* .

*2015*). Since *IR60b-GAL4* appears to be a weaker driver than these pharyngeal *Gr* drivers, we found it necessary to devise a new preparation that images IR60b cell bodies directly in the pharynx. Briefly, we removed the head and then excised the labellum from it such that the pharynx remained intact and accessible in the head. We then mounted the head on a slide such that a fluid stimulus could be perfused into the sample and make contact with the pharynx (*Figure 7A*). We then used *UAS-GCaMP6s* to image cell bodies (*Chen et al., 2013*). We took special care to avoid misinterpreting fluorescent signals that can arise from shifts in the z-axis during imaging (see Materials and methods).

We found that a sucrose stimulus elicited an increase in fluorescence in IR60b neurons, indicating an increase in $Ca^{2+}$ levels (*Figure 7B*). A control water stimulus did not elicit such an increase, indicating that the response is not due to mechanosensory stimulation (*Figure 7C*). Once the stimulus perfuses through the system to the neuron and the $Ca^{2+}$ response becomes detectable, it takes on the order of 30 s to reach peak amplitude (*Figure 7D,E*). This rise time is comparable to that observed for some tastants in pharyngeal neurons of the larval taste system (*Apostolopoulou et al., 2016*; *Choi et al., 2016*; *van Giesen et al., 2016b*). However, the rise time of the Gr43a neuron, once it begins to respond to sugar, is faster (*LeDue et al., 2015*), suggesting that the Gr43a neuron activates a circuit that stimulates feeding before the IR60b neuron activates a circuit that inhibits feeding.

Our previous behavioral analysis indicated that the IR60b receptor was required for response to sucrose, but not to other stimuli tested, except for 900 mM glucose (*Figure 6A–C*). Consistent with these results, of five stimuli tested in parallel, the IR60b neuron was activated only by sucrose and glucose, each tested at 900 mM (*Figure 7F,G*).

To independently confirm the response of the IR60b neuron to sucrose, we drove the GCaMP6s reporter with a different driver, *IR94f-GAL4,* which is coexpressed in the same neuron (*Figure 2J–L*). Again, we found an increase in fluorescence, with comparable dynamics (*Figure 7—figure supplement 1*).

Finally, we asked whether the sucrose response of the IR60b neuron was dependent on IR60b. We used *IR60b-GAL4* to express GCaMP6s in an *IR60b[1]* background and, in parallel, in a control background. Both these genotypes contained a single copy of the *IR60b-GAL4* driver, whereas the flies in our earlier imaging experiments contained two copies and thus presumably higher levels of GCaMP6s. Nonetheless, sucrose elicited a clear $Ca^{2+}$ signal in the control background. The signal was essentially eliminated in the *IR60b[1]* mutant background (*Figure 7H*).

The simplest interpretation of these imaging results is that sucrose acts via the IR60b receptor to activate the IR60b neuron, which in turn activates a circuit that inhibits feeding.

## The IR60b receptor limits sucrose consumption in freely moving animals

We have shown that the IR60b receptor limits sucrose consumption in a pharyngeal pumping paradigm in which flies are immobilized. We finally asked whether the receptor also limits sucrose ingestion in freely moving flies. To address this question, we used the Fly Liquid-Food Interaction Counter

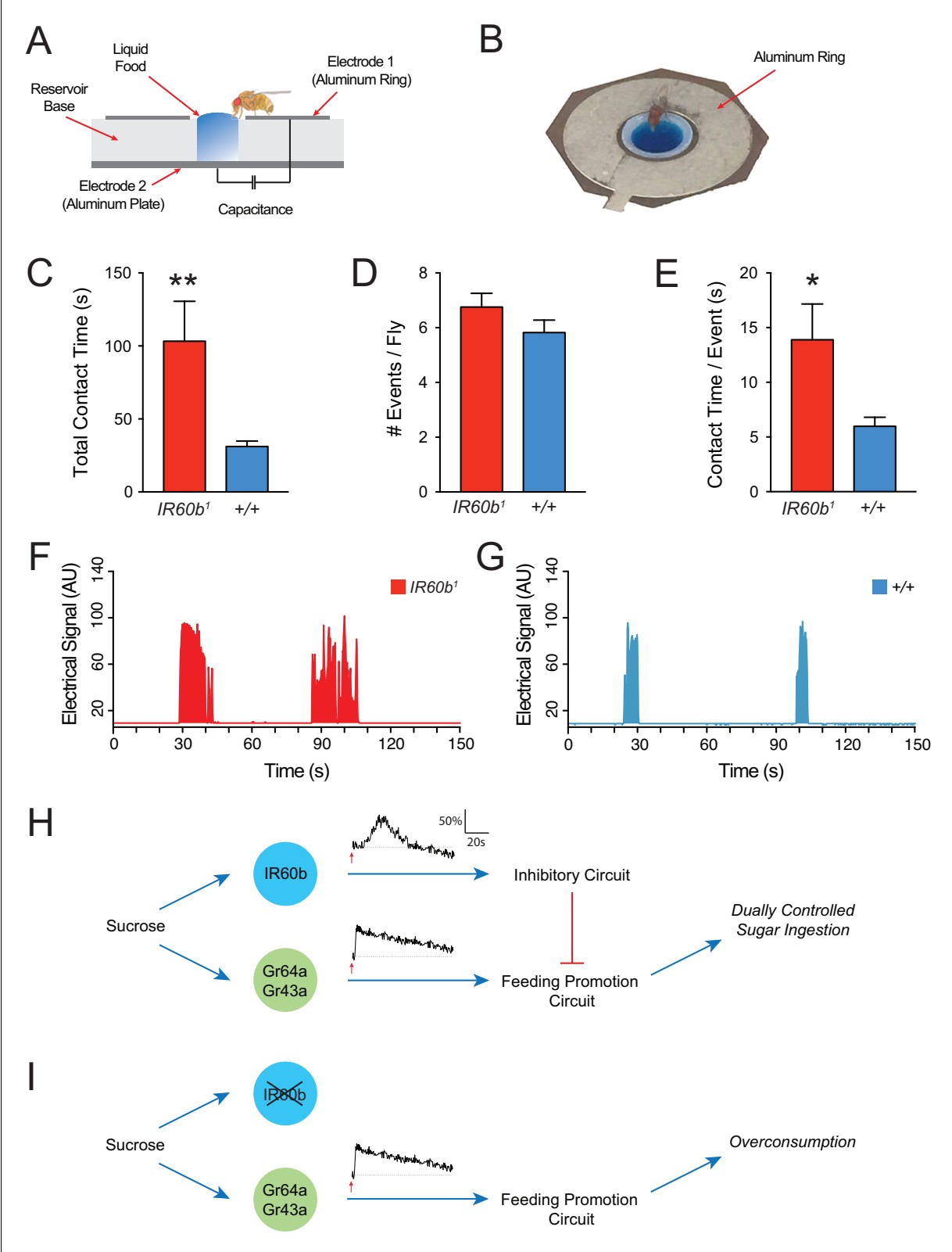

**Figure 8.** The IR60b receptor limits sucrose consumption in freely moving animals. (**A**) The FLIC assay. Diagram has been adapted from the illustration depicted in *Figure 1B* from *Itskov et al. (2014)*. (**B**) Fly contacting sucrose solution in a chamber of the FLIC apparatus. (**C**) Total contact time, (**D**) number of contacts with liquid food per fly, (**E**) contact time per contact event, for *IR60b[1]/IR60b[1]* and control *w Canton S (+/+)*. **p<0.01, *p<0.05, Student's two-tailed t-test, n = 44–46. (**F,G**) Examples of contact events of mutant and control flies. The durations of contact events are longer in the
*Figure 8 continued on next page*

*Figure 8 continued*

mutant. Mated females in (**A–G**) were starved for 12 hr using the same protocol as in the pharyngeal pumping assays. Results were analyzed using custom software code (*Tam, 2017*). (**H**) A dual-control model for regulation of short-term sucrose consumption by pharyngeal neurons. After the fly encounters an acceptable food source, the Gr64a/Gr43a neuron would be activated and promote a feeding response. The IR60b neuron would be activated subsequently and activate a circuit that inhibits the Gr64a/Gr43a neuron, thereby limiting feeding. The absence of IR60b would lead to overconsumption (**I**).

(FLIC) (*Itskov et al., 2014*; *Ro et al., 2014*), and we wrote custom software to analyze the data (Materials and methods).

In the FLIC system, liquid food is placed in a well surrounded by an aluminum ring (*Figure 8A,B*). To access the food, a fly must stand on the ring, which acts as an electrode. When the fly makes contact with the liquid, it closes an electrical circuit between the aluminum ring, the liquid food, and an aluminum plate at the base of the well, which also functions as an electrode. Thus, when the fly makes contact with the liquid food, an electrical signal is generated. Previous studies have shown that such electrical signals provide a reliable indication of ingestion (*Itskov et al., 2014*).

Freely moving *IR60b*[1] flies showed hallmarks of overconsumption. We analyzed the response to 300 mM sucrose for 15 min, before post-ingestive effects modulate hunger state and feeding behavior. Strikingly, *IR60b*[1] mutants made contact with sucrose for three times longer than controls during this period (*Figure 8C*). We note that in the pharyngeal pumping assay, mutants fed for longer than controls by a comparable factor (three-fold). Does the longer total contact time of mutants reflect a greater number of feeding bouts or a greater mean duration of individual feeding bouts? *IR60b*[1] mutants approached the wells and made contact with the food the same number of times as the control (*Figure 8D*), as if locomotion and foraging capabilities are unaffected by the mutation. However, the mean duration of individual contact events was longer in the *IR60b*[1] (*Figure 8E–G*), thereby accounting for the greater overall contact time.

The simplest interpretation of these results is that the IR60b receptor is required not only to prevent overconsumption of sucrose in the pharyngeal pumping paradigm but also in freely moving flies under conditions more comparable to natural foraging and feeding. IR60b acts to limit the duration of individual feeding bouts.

## Discussion

A great deal of attention has been paid to sugar-sensing neurons that are housed in gustatory organs and that promote feeding. Here, we identify within a taste organ a sugar-sensing neuron that inhibits feeding, and we identify a receptor that underlies this inhibition. The neuron and the receptor provide the cellular and molecular underpinnings of a new element in the circuit logic of feeding regulation.

### A new role for an ionotropic receptor in an orphan neuron

This study maps IR60b to a previously undefined pharyngeal neuron. The study identifies a role for both the neuron and the receptor, which belongs to the *IR20a* clade of IRs. Only two members of this clade have previously been functionally analyzed, and both were implicated in mate-sensing (*Koh et al., 2014*). The current study extends the functions of this clade to include sugar-sensing, which has not previously been observed for any member of the entire IR family.

The simplest interpretation of our results is that IR60b is a sucrose receptor or co-receptor. IR60b maps to a pharyngeal neuron that does not express any previously defined Gr sugar receptors such as Gr43a and Gr64a (*LeDue et al., 2015*). Although we have not identified any Gr receptors that map to the IR60b neuron, both IR94f and IR94h co-mapped with IR60b. Further studies will be required to determine if either of these other IRs form a complex with IR60b. We note that of the *IR* genes of this clade analyzed, *IR94h* has the next lowest DoS value after *IR60b* (*Figure 1D*) (*Koh et al., 2014*).

Among a wide variety of other stimuli tested, only glucose at the highest tested concentration appears to elicit a response via IR60b. Sucrose is a dissaccharide composed of the monosaccharides

glucose and fructose, and it is possible that at high concentrations glucose may bind to a site that has evolved to bind sucrose.

We note that the high specificity of the IR60b phenotype argues against a role for the IR60b receptor in signaling osmolarity, viscosity, or mechanosensory stimulation.

## A new element in the logic of feeding control

Fifty years of research on fly taste has enshrined the orthodoxy that sugars act on taste organs to stimulate feeding, whereas bitter compounds act on them to inhibit feeding. In this study, we have identified a peripheral taste neuron that detects sucrose but that inhibits feeding. That is, sucrose assumes a negative rather than a positive valence with respect to this neuron.

We note that both loss-of-function and gain-of-function experiments suggest that the IR60b neuron does not alone dictate the level of consumption. In the absence of *IR60b,* the fly does not feed continuously, and when the IR60b neuron is activated optogenetically the fly does not terminate all feeding. Rather, the IR60b neuron appears to act as a brake, to temper the activity of other neurons that promote feeding. These other neurons include the pharyngeal neurons that coexpress Gr sugar receptors (*Figure 2P*)(*LeDue et al., 2015*).

Sucrose has high-caloric value. Why after detecting and locating a source of sucrose would it be advantageous for a fly to have a taste neuron that inhibits its consumption? First, although sucrose is an energy-rich nutrient, a large increase in sucrose levels may disrupt the nutrient homeostasis in the fly (*Miyamoto et al., 2012*). The IR60b neuron may provide a mechanism for moderating the rate of sucrose intake and thereby helping to maintain homeostasis.

There may be special advantages to activating an inhibitory circuit from a peripheral taste receptor. The inhibition can begin quickly, in contrast to the post-ingestive inhibition that occurs only after sugars reach nutrient sensors in the CNS (*Dus et al., 2015*; *Miyamoto et al., 2012*). A fast-acting inhibition could prevent a rapid influx of sugar into the system. By tempering the dynamics of sugar intake, the circuit could prevent the system from quickly moving far from equilibrium and requiring major metabolic costs to restore. A similar mechanism could exist in mammals to limit water ingestion: peripheral osmolarity detectors may be responsible in part for the cessation of water drinking before systemic osmolarity levels are affected (*Bourque, 2008*).

In a more general sense, there are advantages to a system that has circuit mechanisms of both activation and inhibition. Dual control may allow the system to be fine-tuned more precisely. For example, changes in the internal state of the fly could in principle affect feeding by modulation of both positive and negative control mechanisms. Positive control mechanisms in this system have recently been identified (*LeDue et al., 2015*; *Yapici et al., 2016*), and now IR60b neurons identify a negative control mechanism; together, these mechanisms may allow for a more precise orchestration of feeding behavior. For example, satiety negatively modulates Gr43a neurons, and it will be interesting to determine whether satiety positively modulates IR60b neurons.

Flies encounter many sugars in their environment (*Das et al., 2016*). Why might it be advantageous to have a mechanism that is tuned to sucrose as opposed to other sugars? As fruits ripen, sucrose levels decline and the levels of other sugars increase in many cases (*Akazawa and Okamoto, 1980*; *Chareoansiri and Kongkachuichai, 2009*; *Hardy, 1967*; *Schwieterman et al., 2014*). *Drosophila melanogaster* prefers feeding on fruits that are at advanced stages of ripening. Perhaps a circuit that inhibits feeding on sources rich in sucrose might tend to favor feeding on food sources with low sucrose concentrations. Such food sources might be richer in monosaccharides or other nutrients, or in microbial species to which *Drosophila melanogaster* is well adapted.

How is this new circuit element integrated into the overall circuit logic of feeding regulation? One possibility is that under natural conditions the dynamics of the IR60b response are slower than those of the pharyngeal Gr43a neuron, a neuron that promotes feeding (*LeDue et al., 2015*; *Yapici et al., 2016*). According to this model, after the fly encounters an acceptable food source, the Gr43a neuron would be activated and promote a feeding response. The IR60b neuron would be excited subsequently and activate a circuit that inhibits the Gr43a neuron, thereby limiting feeding (*Figure 8H*). The absence of IR60b would lead to overconsumption (*Figure 8I*).

We note finally that the central projections of the IR60b neuron differ from those of other pharyngeal neurons. *GAL4* drivers representing receptors of the *IR20a* clade label projections in the dorsoanterior subesophageal zone (*Koh et al., 2014*). However, drivers of *IR60b* and *IR94f*, which are coexpressed (*Figure 2P*), differ from the others in that they show no projections toward the midline

(*Koh et al., 2014*). In this respect, the *IR60b* driver also differs from *Gr-GAL4* drivers expressed in the pharynx (*Kwon et al., 2014*). A detailed study using GRASP analysis (*Feinberg et al., 2008*) to identify synaptic partners of *IR60b* neurons may help identify higher-order neural circuits that the IR60b neuron connects to, and may determine whether the distinct projections of the IR60b neuron underlie its distinct function.

### Implications for other functions and other systems

Mutants of the IR60b receptor had a highly specific phenotype when tested with a broad panel of tastants, suggesting that IR60b acts specifically to prevent overconsumption of sucrose. It will be interesting to determine by mutational analysis whether either IR94f or IR94h act in the same neuron to limit overconsumption of other tastants not considered in the present analysis. Likewise, it will be interesting to determine if other taste neurons in the fly play analogous roles, perhaps limiting over-consumption of other taste stimuli.

It is striking that sucrose has a negative valence that is mediated via IR60b neurons but a positive valence via the Gr43a neuron. In both flies and mammals, NaCl has a negative valence with some taste neurons (*Amrein and Thorne, 2005*; *Hiroi et al., 2004*), and a positive valence with others (*Zhang et al., 2013*). However, NaCl acts on these different neurons at different concentrations, whereas sucrose acts on the IR60b and Gr43a neurons at the same concentration.

Finally, it will be interesting to determine if there are taste neurons in mammals that detect sugars and limit their consumption by a logic similar to that identified in this study. Overconsumption of sugar is a major cause of a rapidly expanding worldwide obesity epidemic in humans (*WHO, 2011*). The identification of neurons that limit sugar consumption could provide targets useful in controlling this massive global health problem.

## Materials and methods

### Fly strains

Flies were reared on standard cornmeal-dextrose agar food at 25°C and 40% humidity. *IR-GAL4* drivers are described in *Koh et al. (2014)*. *Gr-GAL4* drivers are described in *Weiss et al. (2011)*. The *ppk28-LexA* transgene was originally described in *Thistle et al. (2012)*. The *UAS-mCD8-GFP* and *LexAop-mtdTomato* constructs are described in *Koh et al. (2014)*. *UAS-tetanus toxin (TNT)* line was generated from second and third chromosome insertion lines originally described in *Sweeney et al. (1995)*. *UAS-CsChrimson* was a generous gift from Vivek Jayaraman. *UAS-GCaMP6s* was a gift from Douglas Kim. All lines were backcrossed for at least five generations to $w^{1118}$ Canton-S before testing in behavioral assays.

### *IR60b* deletion

CRISPR-Cas9 homologous recombination was employed to generate the $IR60b^1$ and $IR60b^2$ mutations. Guide chiRNAs were selected using the CRISPR Optimal Target Finder resource on the fly-CRISPR website (*Gratz et al., 2014*). Gibson Assembly was used to generate guide chiRNA using the Gibson Assembly Master Mix (New England BioLabs, Inc: Ipswich, MA).

*Drosophila* embryos were injected with guide chiRNA and donor plasmids by Bestgene, Inc. (Chino Hills, CA). Two non-sibling alleles were isolated by screening *DsRed*, and were subsequently back-crossed to the $w^{1118}$ CantonS stock for five generations before experimentation. DNA sequence analysis determined that the mutant allele lacks ~66% of the *IR60b* coding region. Detailed description of guide chiRNA sequences, verification primers, and mutation strategy is provided in *Supplementary file 1*.

### Expression analysis

To perform cell-counting experiments, *GAL4* lines were used to express *UAS-mCD8::GFP* (*Lee and Luo, 1999*). Lines containing *IR60b-GAL4* had two copies of the transgene to compensate for low *IR60b-GAL4* expression levels, while partner *IR-GAL4*, *Gr-GAL4*, or *ppk28-GAL4* transgenes were present as a single copy. *UAS-mCD8::GFP* was present as a single copy in all tested lines. Flies were aged between 10 and 35 days to ensure sufficient GFP production for visualization. Female flies

were decapitated and mounted on a microscope slide in 50% glycerol for immediate imaging on a Zeiss 510 Meta confocal microscope.

We systematically mapped the expression of *GAL4* drivers by analyzing z-stacks to count the number of cell bodies expressing a *UAS-mCD8-GFP* reporter in the LSO. For each genotype, n $\geq$ 10 flies were examined.

### Tastant panel

Tastants from J.T. Baker (Central Valley, PA), American Bio, Inc (Natick, MA), and Sigma-Aldrich (St. Louis, MA) were obtained at high purity. Samples were stored as aliquots at −20°C long-term and kept at 4°C during experiments. Thawed aliquots were discarded after one week.

### Pharyngeal pumping assay

We employed a pharyngeal pumping assay, modified from that of *Manzo et al. (2012)*. Mated females were aged for 10–14 days to allow for tetanus toxin (TNT) or Chrimson (CHR) to accumulate via expression from the relatively weak *IR60b-GAL4* driver. Flies were also aged for 10–14 days in *IR60b[1]*, *IR60b[2]*, and *IR94f-GAL4* experiments. Before testing, flies were placed in a 1000-µl pipette tip. A second 1000-µl pipette tip was inserted into the first tip so as to contain the fly for a starvation period of 12–14 hr. Flies were placed in a 100-mm Petri dish with three Kimwipes wetted with 5 ml water, and then transferred to a humidity-controlled room to prevent dehydration.

After starvation, the fly was aspirated into another 1000-µl pipette tip such that it was immobilized with its head and proboscis exposed. Occasionally, ends of the 1000-µl pipette tips were trimmed with a razor blade to widen the opening for the fly's head. Flies were mounted on a micromanipulator for video recording under a Nikon SMZ800 microscope with an attached Sony HD Camcorder. Before food was delivered, the fly was presented a water droplet to ensure the animal was not overly desiccated from starvation. If it consumed water for longer than 10 s, the animal was discarded. Only ~5% of flies were discarded by this criterion.

We added 0.4 µg/µl erioglaucine blue dye to the liquid food to facilitate data acquisition. Food was presented with a P20 PipetteMan mounted on a micromanipulator, allowing for fine adjustments during delivery. The fly was offered food for 2 s. Flies that ingested liquid were allowed to continue feeding until they freely terminated feeding. After breaking contact with the drop, flies were given 3 s of rest before a subsequent presentation, in which they were given another 2 s to initiate a second bout. This process was repeated until the fly no longer responded to food. Typically, 2 s is sufficient presentation time to initiate feeding, as longer presentations did not increase the likelihood of initiating a feeding bout. On average, flies engaged in 1–3 bouts. If a fly did not initiate feeding after four attempted presentations, it was discarded as a non-responder. When testing different concentrations and tastants, the fraction of non-responders did not differ significantly between control, mutant, or transgenically manipulated flies, indicating there were no differences in the capacity to initiate feeding between the different genotypes. Typically, the fraction of non-responders was less than ~10% for appetitive sugars like sucrose.

Each fly was used for only one experiment to prevent previous experience from influencing its responses. The investigator was blind to the genotypes of the flies. The video was analyzed using QuickTime Media Player. Within each experiment, responses of different genotypes were measured in parallel, at the same time of day, by the same investigator, and within the same 3–5 day time window.

### Optogenetic activation

Experiments with *UAS-CsChrimson* were performed as described for the pharyngeal pumping assay, except that flies were kept in the dark prior to testing, including a 24-hr period during which flies were given cornmeal food supplemented with 0.5 mM all-trans-retinal, followed by a 12-hr starvation period. After being presented a water droplet, the whole fly was illuminated by high-intensity 626 nm red LEDs at an intensity of ~5 W/m$^2$ for the duration of the assay.

### Calculation of ingested volumes

The volumes ingested by single flies (*Figures 3F* and *5C*) were measured as follows: 10 serial two-fold dilutions of 0.4 µg/µl erioglaucine were prepared for spectrophotometer analysis using a

NanoDrop 2000c Spectrometer (Thermo Scientific: Wilmington, DE). The average optical density (OD$_{630}$) was determined for 1 μl of each dilution (n ≥ 3 for each concentration). Average ODs were then used to plot a standard line for absorbance of blue dye (OD) versus the dye concentration. The resulting slope was 0.14 ± 0.001 OD/μl, and was used as a factor for converting OD values of 0.4 μg/μl erioglaucine to volume of fluid ingested by the fly.

A single mated female was fed 900 mM sucrose with 0.4 μg/μl erioglaucine blue dye in the pharyngeal pumping assay. After feeding, the fly was homogenized in water and samples were centrifuged for 1 min at 14,000 RPM to pellet the cuticle. 1 μl of supernatant from homogenized flies were then assayed with the NanoDrop Spectrometer, and an average OD value was used to determine the amount of blue dye present in each fly (n ≥ 4 OD values were averaged for each fly). The slope of the standard line for 0.4 μg/μl erioglaucine was then used to convert average OD values into volumes ingested. To control for endogenous absorbance of the extract, four females were fed 900 mM sucrose without blue dye, and OD values for each fly were obtained. The calculated values for these control flies were averaged, yielding a standard absorbance value in the OD$_{630}$ range. This standard value was subtracted from each measurement from the flies that were fed 900 mM sucrose with 0.4 μg/μl erioglaucine, before determining the final calculated ingested volumes.

## Calcium imaging

To image the cell bodies of IR60b neurons directly in the pharynx, we first removed the heads of mated females. To increase access to the esophagus and pharyngeal sensilla, the labellum lobes were carefully excised while leaving the rest of the proboscis and pharynx intact. Heads were mounted in water on a slide with three 18 × 18 mm bridging slips, positioned such that two gaps were left between the bridging slips, so liquid stimulant could be perfused through the sample (*Figure 7A*). A minimal amount of water was used to minimize dilution of the stimulus during the perfusion process. The head and bridging slips were then secured with a 22 × 40 mm coverslip, positioned with roughly 10 mm of overhang from the microscope slide, which would later allow for liquid stimulant to be delivered to the preparation when the sample was inverted. The three non-overhanging sides of the coverslip were secured with nail polish.

*UAS-GCaMP6s* fluorescence was viewed with an inverted Zeiss 510 Meta confocal microscope. The IR60b neurons were visualized with a 20X objective, with a digital zoom of 2.5. Images were acquired such that a single 512 × 512 resolution frame was scanned each second, with single-line averaging. The pinhole was opened to 3.42 Airy Units (7.3 μm), with the 488 nm laser at 15%. Stimulus was perfused into the sample by delivering tastant solution to the overhanging portion of the coverslip, using a P20 PipetteMan. Changes in fluorescent activity were recorded for 5 m after delivery of the stimulus.

Scanned images were binned into 30 s intervals, and a maximum change in fluorescence (ΔF/F) was calculated for each bin in order to generate a response curve as depicted in *Figure 7*. ΔF/F for each binned time interval was calculated as the maximum fluorescence change divided by the average baseline fluorescence of the 10 consecutive frames taken immediately before stimulation. Fluorescent intensities were obtained by using open-source Fiji/ImageJ software (https://fiji.sc) to draw a region-of-interest (ROI) around cell bodies, and measuring an average pixel intensity for the ROI in each frame using the Time-Series Analyzer Plugin written by Balaji, J. (https://imagej.nih.gov/ij/plugins/time-series.html).

When imaging single cell bodies, false positives can occur if the preparation shifts out of the initial plane of focus, along the z-axis, thereby causing changes in the baseline fluorescence levels during the assay. To prevent false positives, care was taken to ensure that cell bodies remained in the correct plane of focus. Before the stimulus, we selected focal landmarks in both the DIC and green fluorescent channels, and closely monitored them during the experiment. If the sample shifted, it was refocused to the reference landmarks, and the out-of-focus frames were excluded from the ΔF/F calculations. Typically, refocusing took 5–10 s, which is less than the binned 30 s time segments used in ΔF/F calculations. Thus, we recorded sufficient information to measure the maximum ΔF/F for each 30 s time interval depicted in *Figure 7E–G*.

## FLIC assay

The FLIC *Drosophila* Behavior System was from Sable Systems International (North Las Vegas, NV) and was described by *Ro et al. (2014)*. The FLIC Monitor Software (version 2.1, downloaded from wikiflic.com) was used to collect raw data from the Drosophila Feeding Monitor (DFM).

Before the assay, mated females were starved in 1000-µl pipette tips using the same protocol as for the pharyngeal pumping assay. After loading 900 mM sucrose into the DFM, the FLIC Monitor Software was initialized. Six mutant and six control flies were then quickly aspirated into the FLIC system (12 single-well setup). Typically, loading took no longer than 2 min. We analyzed the first 15 m of feeding, a period chosen to reduce the influence of post-ingestive effects that modulate hunger state and feeding behavior.

To automate the analysis of raw FLIC data, we wrote custom software in the R programming language (*Tam, 2017*). Feeding events were detected as peaks in raw FLIC traces using the open-source numerical analysis R package 'pracma' (available at https://cran.r-project.org/web/packages/pracma/index.html). Thresholds and parameters of the software are user-customizable. To assess the accuracy of our software, we selected 10 FLIC traces randomly for manual ground-truth scoring. Our algorithm correctly identified 97.6% of the peaks with a false-positive rate of 1.8%. Source code for the software can be obtained at https://github.com/edrictam/FLIC-analysis (with a copy archived at https://github.com/elifesciences-publications/FLIC-Analysis).

## Statistical analysis

All statistical tests described in the figure legends were performed using GraphPad Prism 7. D'Agostino-Pearson and Kolmogorov-Smirnov normality tests were employed to confirm that our data approximately followed Gaussian distributions, thereby allowing the use of parametric analysis.

## Acknowledgements

We thank A Dahanukar, D Chen, and members of the Carlson lab for comments on the ms. We thank HK Dweck for help setting up the FLIC hardware. We thank J Belina for help with the CsChrimson assay. We thank the Bloomington *Drosophila* Stock Center for fly lines. pDsRed-attP was a gift from M Harrison and K O'Connor-Giles and J Wildonger (Addgene plasmid # 51019, unpublished), as was pU6-BbsI-chiRNA (Addgene plasmid # 45946). We especially thank Marek Chodakowski, whose outstanding technical support has been essential to the lab for many years.

RMJ was supported by an NIH NRSA (5F32DC013507). JSS was supported by an NSF Graduate Research Fellowship, NIH T32 GM007499, and the Dwight N and Noyes D Clark Scholarship Fund. The project was supported by NIH grants to JRC.

## Additional information

### Funding

| Funder | Grant reference number | Author |
| --- | --- | --- |
| National Institutes of Health | 5F32DC013507 | Ryan M Joseph |
| National Institutes of Health | T32 GM007499 | Jennifer S Sun |
| National Science Foundation | Graduate Research Fellowship | Jennifer S Sun |
| National Institutes of Health | | John R Carlson |

The funders had no role in study design, data collection and interpretation, or the decision to submit the work for publication.

### Author contributions

RMJ, Conceptualization, Resources, Formal analysis, Funding acquisition, Investigation, Methodology, Writing—original draft, Writing—review and editing; JSS, Resources, Funding acquisition, Validation, Writing—review and editing; ET, Software, Formal analysis, Validation, Writing—review and

editing; JRC, Conceptualization, Supervision, Funding acquisition, Methodology, Writing—original draft, Writing—review and editing

Author ORCIDs
Ryan M Joseph, http://orcid.org/0000-0001-8203-7154
Jennifer S Sun, http://orcid.org/0000-0002-4274-0504
John R Carlson, http://orcid.org/0000-0002-0244-5180

## Additional files

### Supplementary files

• Supplementary file 1. Sequences for various primers used in CRISPR generation of *IR60b* mutant. (A) cloning primers generated for the CRISPR guide chiRNA, (B) CRISPR donor plasmid, and (C) screening to verify the removal of *IR60b* coding regions. In (A), lowercase nucleotides denote primer sequences homologous to the Gibson Assembly plasmid regions.

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
