## [Decision Letter]

Thank you for submitting your article "A receptor and neuron that activate a circuit limiting sucrose consumption" for consideration by *eLife*. Your article has been favorably evaluated by Eve Marder (Senior Editor) and three reviewers, one of whom, Ronald L Calabrese (Reviewer #1), is a member of our Board of Reviewing Editors. The following individuals involved in review of your submission have agreed to reveal their identity: Paul A Garrity (Reviewer #2); Craig Montell (Reviewer #3).

The reviewers have discussed the reviews with one another and the Reviewing Editor has drafted this decision to help you prepare a revised submission.

Summary:

This is a very interesting manuscript that describes a new taste neuron and associated putative molecular receptor, IR60b, with a strong specificity for sucrose in *Drosophila*. The experiments show convincingly that this neuron/receptor combo acts to limit sucrose consumption in the flies. The neuron when activated optogenetically limits sugar consumption of a sugar that does not activate the neuron. When the neuron or receptor are inactivated then the fly consumes more than the wildtype amount of sucrose. Experiments in freely moving flies show that inactivation of the receptor (loss of function mutation) causes longer duration feeding bouts. Ca imaging experiments show that sucrose activates the neuron and that the dynamics of the activation are slow. Because the activation of neurons/receptors that promote sucrose feeding are more rapid, the authors propose a push pull model where rapidly activating neurons that convey the positive valence of sucrose for promoting feeding are gradually inhibited by the slowly activating neuron/receptor that conveys a negative valence for sucrose to limit feeding.

Essential revisions:

The paper is well written and clearly illustrated, and presents all necessary data without clutter by using figure supplements, although there might be some consolidation of figures. The experiments appear carefully controlled and appropriate statistical methods are used. The findings are novel and should be of wide interest to the feeding and sensory community.

1) The authors used CsChrimson to activate the Ir60b neurons. However, rather than testing the effects on sucrose consumption they used trehalose, which they justified by stating that "trehalose elicits a substantial level of feeding, and whose consumption was not affected by silencing of IR60b activity." However, the effects of activation on sucrose consumption should still be presented as this is central to their paper. We strongly suggest that the authors perform this experiments, which should be rather quick given that all fly lines needed are still available.

2) Provide an explanation as to why the feeding times are so different between figures, even when using the same genotype? Examples include *UAS-TNT*/+ for sucrose feeding (Figure 4 and Figure 4—figure supplement 1) and IR60b/+ for trehalose feeding (Figure 4 for IR60b/+)? If the experimental results vary considerably from day to day or with different investigators, indicate whether or not all of the data in a given panel were performed at the same time by the same person.

3) To explain the existence of such an inhibitory sugar neuron, the authors state in the Discussion that, "The IR60b neuron may provide a mechanism for moderating the rate of sucrose intake and thereby helping to maintain homeostasis." Along these lines, is the effect of activating these neurons on behavior affected by the state of satiation? It should also be feasible to test whether the level of activation of these neurons (using GCaMP6s) is affected by the state of satiation. If these or similar experiments have been performed they should be reported, otherwise Discussion should be altered to include discussion of how satiation might be expected to affect the responses of the IR60b cell system.

4) In Abstract (similarly in other sections of the paper) the authors say "…we identify a taste neuron that is necessary and sufficient to limit sucrose consumption in *Drosophila*." We understand how they arrive at sufficiency but they are over simplifying in their claim of necessity. Sucrose consumption is larger in flies where the neuron/receptor is inactivated, *but* sucrose consumption is nevertheless limited albeit it at a higher amount. Here and throughout the manuscript, it seems to be important to make this distinction and not convey a simplistic impression to readers, who might go for the punch line and not dig deeper. Some qualifying phrase like 'to normal wild type levels' seems in order for each claim of necessity. Moreover, the statements in the Abstract and Introduction implying the relevance these findings to human sugar consumption and obesity seem too strong. Given the dramatic differences in anatomy between the systems along with the paucity of information about how the IR60b cell system operates in *Drosophila*, it is not compelling and perhaps best left to the Discussion. The paper is interesting enough without it.

5) The text is sometimes phrased as though IR60b = sucrose receptor (as in the Abstract: "IR60b shows a high degree of chemical specificity when tested with a broad panel of tastants"). It is not formally established whether IR60b directly participates in the formation of the sucrose receptor or might have some other role in the neuron that is necessary for the function of the receptor. Also, they observe two other IRs expressed in the neuron, but don't test whether these IRs are also required for sucrose detection.

6) The notion that these neurons can serve as a brake on consumption is demonstrated, but its significance is never really explored. One can imagine at least two roles of such a receptor. On the one hand, this neuron might provide a level of feeding control that responds to physiological state (becoming less active as the animal becomes sated). Another possibility is that it is not so much about controlling feeding but rather about shifting flies toward foods that have less sucrose (and perhaps are richer in monosaccharides). [In Discussion, there is a paragraph "Flies encounter many sugars in their environment (Das et al. 2016). Why might it be advantageous to have a mechanism that is tuned to sucrose as opposed to other sugars? As fruits ripen, sucrose levels decline and the levels of other sugars increase in many cases (Akazawa and Okamoto 1980, Chareoansiri and Kongkachuichai 2009, Hardy 1967, Schwieterman et al. 2014). *Drosophila melanogaster* prefers feeding on fruits that are at advanced stages of ripening. Perhaps a circuit that inhibits feeding on sources rich in sucrose might tend to favor feeding on food sources with low sucrose concentrations."] These are interesting issues, but are not explored experimentally. At least they could be explored more fully in Discussion; Why go for sucrose in the first place (positive valence sensory neurons)? What might be the disadvantage of feeding on sucrose rich fruit?

7) In Abstract (similarly in other sections of the paper) the authors say "…we identify a taste neuron that is necessary and sufficient to limit sucrose consumption in *Drosophila*." We understand how they arrive at sufficiency but they are over simplifying in their claim of necessity. Sucrose consumption is larger in flies where the neuron/receptor is inactivated, BUT sucrose consumption is nevertheless limited albeit it at a higher amount. Here and throughout the manuscript, it seems to be important to make this distinction and not convey a simplistic impression to readers, who might go for the punch line and not dig deeper. Some qualifying phrase like 'to normal wild type levels' seems in order for each claim of necessity.

---

## [Author Response]

*Essential revisions:*

*[…] 1) The authors used CsChrimson to activate the Ir60b neurons. However, rather than testing the effects on sucrose consumption they used trehalose, which they justified by stating that "trehalose elicits a substantial level of feeding, and whose consumption was not affected by silencing of IR60b activity." However, the effects of activation on sucrose consumption should still be presented as this is central to their paper. We strongly suggest that the authors perform this experiments, which should be rather quick given that all fly lines needed are still available.*

We have added the requested experiment (Figure 4—figure supplement 2; Figure 4—figure supplement 2 legend), which shows that optogenetic stimulation of the IR60b neuron does not reduce sucrose feeding, and we have provided an explanation as to why this result is not surprising. Sucrose activates the IR60b neuron, and it is likely that this activation is sufficiently strong that red light does not produce a demonstrable incremental effect in the reduction of consumption. The revised manuscript explains in more detail that a more sensitive assay is provided by trehalose, which does not activate the IR60b neuron but elicits feeding (subsection “Activation of the IR60b neuron decreases feeding time”, first paragraph). In the absence of sucrose-induced activation of IR60b, the effect of optogenetic activation is easily discernible (Figure 4).

*2) Provide an explanation as to why the feeding times are so different between figures, even when using the same genotype? Examples include UAS-TNT/+ for sucrose feeding (Figure 4 and Figure 4—figure supplement 1) and IR60b/+ for trehalose feeding (Figure 4 for IR60b/+)? If the experimental results vary considerably from day to day or with different investigators, indicate whether or not all of the data in a given panel were performed at the same time by the same person.*

The reviewer is correct that there is variation in feeding levels, of unknown cause, between experiments. For this reason, we were extremely careful to compare different genotypes in parallel and have indicated this by adding the following statement:

“Within each experiment, responses of different genotypes were measured in parallel, at the same time of day, by the same investigator, and within the same 3-5 day time window.”

*3) To explain the existence of such an inhibitory sugar neuron, the authors state in the Discussion that, "The IR60b neuron may provide a mechanism for moderating the rate of sucrose intake and thereby helping to maintain homeostasis." Along these lines, is the effect of activating these neurons on behavior affected by the state of satiation? It should also be feasible to test whether the level of activation of these neurons (using GCaMP6s) is affected by the state of satiation. If these or similar experiments have been performed they should be reported, otherwise Discussion should be altered to include discussion of how satiation might be expected to affect the responses of the IR60b cell system.*

This is a very interesting question. We have not carried out these experiments, in part because in our preparation the flies are decapitated and the IR60b neurons are not in contact with the body hemolymph, but we have added a statement to the Discussion of how satiety might affect the responses of IR60b neurons (subsection “A new element in the logic of feeding control”, seventh paragraph).

*4) In Abstract (similarly in other sections of the paper) the authors say "…we identify a taste neuron that is necessary and sufficient to limit sucrose consumption in Drosophila." We understand how they arrive at sufficiency but they are over simplifying in their claim of necessity. Sucrose consumption is larger in flies where the neuron/receptor is inactivated, but sucrose consumption is nevertheless limited albeit it at a higher amount. Here and throughout the manuscript, it seems to be important to make this distinction and not convey a simplistic impression to readers, who might go for the punch line and not dig deeper. Some qualifying phrase like 'to normal wild type levels' seems in order for each claim of necessity. Moreover, the statements in the Abstract and Introduction implying the relevance these findings to human sugar consumption and obesity seem too strong. Given the dramatic differences in anatomy between the systems along with the paucity of information about how the IR60b cell system operates in Drosophila, it is not compelling and perhaps best left to the Discussion. The paper is interesting enough without it.*

As requested, we have deleted or qualified the term “necessary” in each case, at various locations in the revised manuscript. Also as requested, we have eliminated the references to human health in the Abstract and Introduction.

*5) The text is sometimes phrased as though IR60b = sucrose receptor (as in the Abstract: "IR60b shows a high degree of chemical specificity when tested with a broad panel of tastants"). It is not formally established whether IR60b directly participates in the formation of the sucrose receptor or might have some other role in the neuron that is necessary for the function of the receptor. Also, they observe two other IRs expressed in the neuron, but don't test whether these IRs are also required for sucrose detection.*

We have changed the phrasing from “IR60b shows a high degree of chemical specificity” to “the *IR60b* phenotype shows a high degree of chemical specificity.”

*6) The notion that these neurons can serve as a brake on consumption is demonstrated, but its significance is never really explored. One can imagine at least two roles of such a receptor. On the one hand, this neuron might provide a level of feeding control that responds to physiological state (becoming less active as the animal becomes sated). Another possibility is that it is not so much about controlling feeding but rather about shifting flies toward foods that have less sucrose (and perhaps are richer in monosaccharides). [In Discussion, there is a paragraph "Flies encounter many sugars in their environment (Das et al. 2016). Why might it be advantageous to have a mechanism that is tuned to sucrose as opposed to other sugars? As fruits ripen, sucrose levels decline and the levels of other sugars increase in many cases (Akazawa and Okamoto 1980, Chareoansiri and Kongkachuichai 2009, Hardy 1967, Schwieterman et al. 2014). Drosophila melanogaster prefers feeding on fruits that are at advanced stages of ripening. Perhaps a circuit that inhibits feeding on sources rich in sucrose might tend to favor feeding on food sources with low sucrose concentrations."] These are interesting issues, but are not explored experimentally. At least they could be explored more fully in Discussion; Why go for sucrose in the first place (positive valence sensory neurons)? What might be the disadvantage of feeding on sucrose rich fruit?*

We agree this is a very interesting question and have added some discussion of this point (subsection “A new element in the logic of feeding control”).

*7) In Abstract (similarly in other sections of the paper) the authors say "…we identify a taste neuron that is necessary and sufficient to limit sucrose consumption in Drosophila." We understand how they arrive at sufficiency but they are over simplifying in their claim of necessity. Sucrose consumption is larger in flies where the neuron/receptor is inactivated, BUT sucrose consumption is nevertheless limited albeit it at a higher amount. Here and throughout the manuscript, it seems to be important to make this distinction and not convey a simplistic impression to readers, who might go for the punch line and not dig deeper. Some qualifying phrase like 'to normal wild type levels' seems in order for each claim of necessity.*

See point 4 above.